# Cation selectivity of the presequence translocase channel Tim23 is crucial for efficient protein import

**Niels Denkert[1†], Alexander Benjamin Schendzielorz[1†], Mariam Barbot[1], Lennart Versemann[1], Frank Richter[1], Peter Rehling[1,2,3]\*, Michael Meinecke[1,3,4]\***

[1]Department of Cellular Biochemistry, University Medical Center Göttingen, Göttingen, Germany; [2]Max Planck Institute for Biophysical Chemistry, Göttingen, Germany; [3]Göttinger Zentrum für Molekulare Biowissenschaften, Göttingen, Germany; [4]European Neuroscience Institute Göttingen, Göttingen, Germany

**Abstract** Virtually all mitochondrial matrix proteins and a considerable number of inner membrane proteins carry a positively charged, N-terminal presequence and are imported by the TIM23 complex (presequence translocase) located in the inner mitochondrial membrane. The voltage-regulated Tim23 channel constitutes the actual protein-import pore wide enough to allow the passage of polypeptides with a secondary structure. In this study, we identify amino acids important for the cation selectivity of Tim23. Structure based mutants show that selectivity is provided by highly conserved, pore-lining amino acids. Mutations of these amino acid residues lead to reduced selectivity properties, reduced protein import capacity and they render the Tim23 channel insensitive to substrates. We thus show that the cation selectivity of the Tim23 channel is a key feature for substrate recognition and efficient protein import.
DOI: https://doi.org/10.7554/eLife.28324.001

\*For correspondence:
peter.rehling@medizin.uni-goettingen.de (PR);
michael.meinecke@med.uni-goettingen.de (MM)

[†]These authors contributed equally to this work

Competing interests: The authors declare that no competing interests exist.

## Introduction

Double membrane bounded mitochondria import over 1000 different proteins synthesized on cytosolic ribosomes (*Endo and Yamano, 2009*; *Neupert and Herrmann, 2007*; *Schmidt et al., 2010*). Different targeting signals direct the proteins into one of the four mitochondrial sub-compartments: outer membrane (OM), intermembrane space (IMS), inner membrane (IM) and matrix. Approximately, 70% of these mitochondrial proteins are synthesized with an N-terminal presequence (*Vögtle et al., 2009*), which directs them across the OM. Once threaded through the OM, the presequence directs preproteins to the presequence translocase (TIM23 complex), located in the inner boundary membrane (*Barbot and Meinecke, 2016*; *Chacinska et al., 2005*). The TIM23 complex transports precursor proteins across the inner membrane, or, if they contain additional sorting signals, inserts them into the IM (*Neupert and Herrmann, 2007*; *van der Laan et al., 2007*). The membrane potential ($\Delta\Psi$) across the energy coupling inner membrane exerts an electrophoretic force on the positively charged presequences, thereby providing energy for the translocation of preproteins. $\Delta\Psi$ is necessary and sufficient for membrane insertion of IM proteins (*van der Laan et al., 2007*), whereas full translocation of proteins into the mitochondrial matrix depends on additional energy provided by the ATP consuming presequence translocase-associated import motor PAM (*Neupert and Brunner, 2002*; *Schendzielorz et al., 2017*). The TIM23 complex consists of the channel forming Tim23 subunit and its homolog Tim17 (*Lohret et al., 1997*; *Maarse et al., 1994*; *Meinecke et al., 2006*; *Ryan et al., 1998*; *Truscott et al., 2001*). Additionally, the receptor protein Tim50 as well as Mgr2 are constitutive subunits of the presequence translocase, whereas Tim21 is specific to the TIM23 complex in the absence of the PAM motor (*Chacinska et al., 2005*; *Ieva et al.,*

**eLife digest** The cells of animals, plants and other eukaryotic organisms contain compartments known as organelles that play many different roles. For example, compartments called mitochondria are responsible for supplying the chemical energy cells need to survive and grow. Two membranes surround each mitochondrion and energy is converted on the surface of the inner one.

Mitochondria contain over 1,000 different proteins, most of which are produced in the main part of the cell and have to be transported into the mitochondria. A transport protein called Tim23 is part of a larger group or 'complex' of proteins that helps to import many other proteins into the mitochondria. This complex sits in the inner membrane, with the Tim23 protein forming a large, water-filled pore through its core that provides a route for proteins to pass through the membrane.

Proteins are made of building blocks called amino acids. The proteins transported by the complex containing Tim23 all have a short chain of amino acids at one end known as an N-terminal presequence. However, it is not clear how the inside of the Tim23 channel identifies and transports this presequence to allow the right proteins to pass through the inner membrane.

Denkert, Schendzielorz et al. studied the normal and mutant versions of a Tim23 channel from yeast to find out which parts of the protein are involved in detecting the N-terminal presequence after it enters the pore. The experiments show that there are several amino acids in Tim23 that play important roles in this process. Furthermore, mitochondria containing mutant Tim23 channels, that are less able to identify the N-terminal presequence, are impaired in their ability to import proteins.

Tim23 proteins in humans and other organisms also contain most or all of the specific amino acids identified in this study, suggesting that the findings of Denkert, Schendzielorz et al. will also apply to other species. Furthermore, the experimental strategy used in this study could be adapted to investigate transport proteins in other cell compartments.

DOI: https://doi.org/10.7554/eLife.28324.002

*2014*). Tim23 was identified as the central pore-forming component of the TIM23 complex by electrophysiological characterization of purified Tim23 as well as patch-clamp analyses of inner membrane derived vesicles, depleted of Tim17 (*Martinez-Caballero et al., 2007*; *Truscott et al., 2001*). Tim23 forms a voltage-activated, water-filled pore with a diameter of 1.3–2.4 nm. To maintain the permeability barrier of the inner membrane it is voltage-regulated by the Tim50 receptor and shows sensitivity towards presequence peptides and full-length preproteins (*Meinecke et al., 2006*; *Truscott et al., 2001*). Many electrophysiological features of purified Tim23, such as voltage-gating, substrate sensitivity and selectivity, were also found in measurements of the TIM23 complex. The role of Tim17 is less clear, though recent studies suggest it might be involved in channel regulation within the complex (*Martinez-Caballero et al., 2007*; *Ramesh et al., 2016*).

Despite its channel dimension, which would allow the simultaneous passage of multiple ions, Tim23 shows a clear preference to conduct cations over anions. Since its discovery this selectivity was speculated to be important to recognize and transport positively charged presequences through the channel. The lack of high-resolution 3D structures on the one hand, and the missing amphipathic character of the predicted transmembrane helices hindered the possibility to construct structure based mutants to investigate the molecular nature and physiological importance of the basic electrophysiological characteristics of the Tim23 channel. In recent years efforts have been made to overcome this issue. Fluorescent mapping has allowed for the first time to show which amino acid residues of the transmembrane helices of Tim23 are likely facing the aqueous channel lumen (*Alder et al., 2008*; *Malhotra et al., 2013*).

In this study, we identify pore-lining amino acids of the Tim23 channel that contribute to ion selectivity. Mutations of these highly conserved amino acids specifically affect the channels selective properties while leaving other electrophysiological characteristics intact. Yeast cells expressing mutant Tim23 channels with decreased selectivity show growth defects and are impaired in the import of mitochondrial proteins. On the protein level, selectivity reduction leads to a highly decreased sensitivity towards substrates. Our data provide evidence for the idea that the biophysical properties of protein-conducting Tim23 channel are essential for its physiological functions.

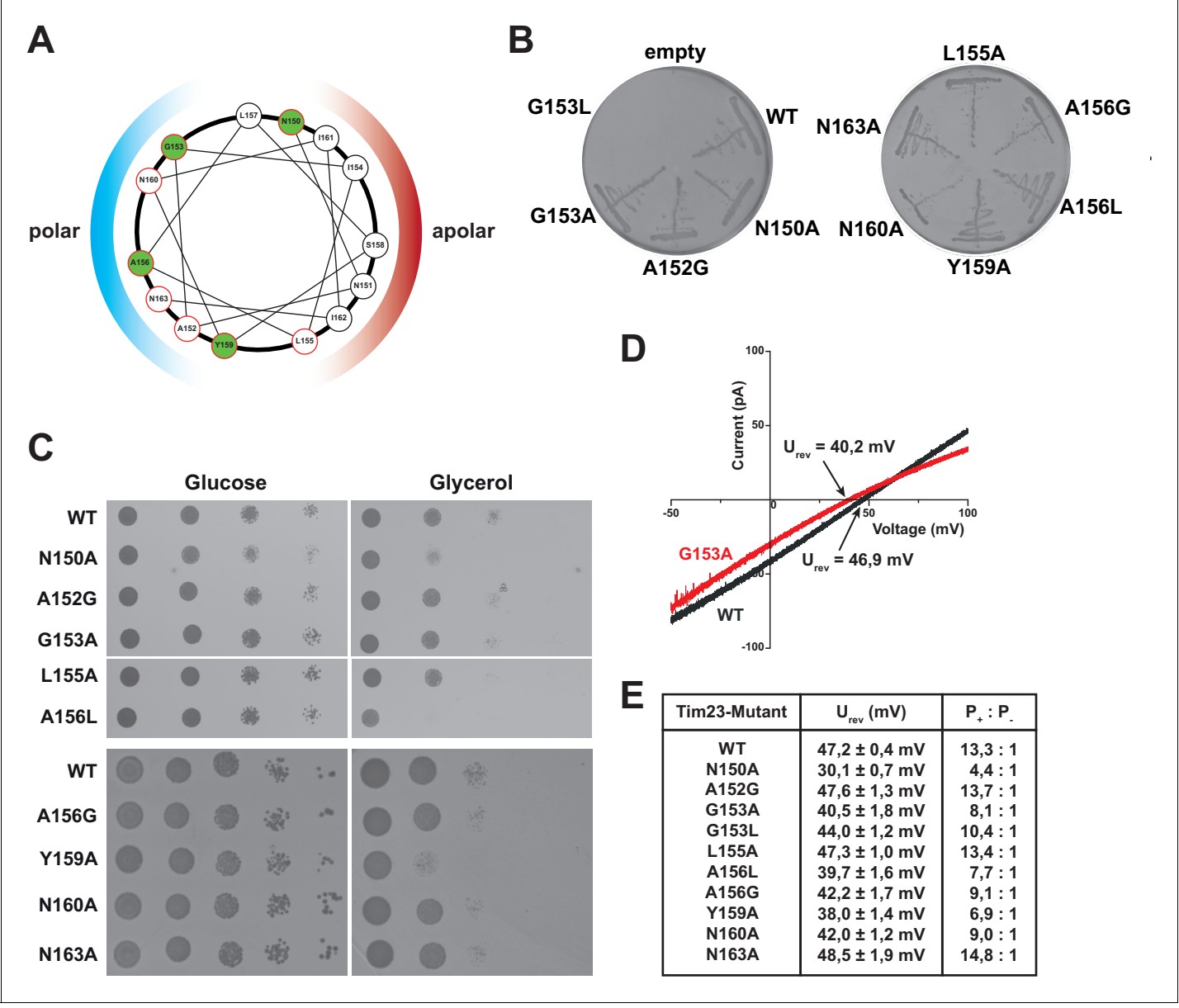

**Figure 1.** Substitutions of pore-lining residues of Tim23 reduce the channel's ion selectivity and lead to a growth defect in *S. cerevisiae*. (A) Helical wheel projection of amino acid residues 150–163 of the second transmembrane helix of Tim23. Highlighted residues in green indicate near 100% conservation. Colored hemispheres indicates polar/apolar facing regions of helix. Residues mutated in this study are circled in red. (B) *S. cerevisiae* strains with chromosomal deletion of the *TIM23* gene, complemented by a plasmid carrying both *URA3* and *TIM23* gene, were transformed with plasmids containing wild type *TIM23* or mutants and tested for viability after plasmid loss on 5-FOA containing medium. (C) *tim23Δ* yeast cells with plasmids containing wild type *TIM23* or pore-lining mutants were grown on fermentable (left) or non-fermentable (right) media at 37°C. Strains WT to A156L were grown on a single plate each for glucose and glycerol respectively. (D) Electrophysiological current-voltage (I–V) curves were recorded at asymmetrical buffer conditions with 12.5-fold KCl gradient for Tim23 (grey) or Tim23$^{G153A}$ (red) to determine reversal potentials. (E) Reversal potentials $U_{rev}$ were experimentally determined for wild type Tim23 and all mutants by independent triplicates at asymmetrical buffer conditions, the ion selectivity was calculated from the mean reversal potential following the Goldman-Hodgkin-Katz equation. Errors represent standard deviation.

DOI: https://doi.org/10.7554/eLife.28324.003

The following figure supplements are available for figure 1:

**Figure supplement 1.** Electrophysiological screening of Tim23 mutants.

DOI: https://doi.org/10.7554/eLife.28324.004

**Figure supplement 2.** Sequence conservation of Tim23.

DOI: https://doi.org/10.7554/eLife.28324.005

## Results

To investigate the physiological function of highly conserved, pore-lining amino acid residues of Tim23 (*Alder et al., 2008*; *Malhotra et al., 2013*) from *Saccharomyces cerevisiae in vivo*, we employed mutants based on substitution of amino acids in the second transmembrane helix (*Figure 1A*). *S. cerevisiae* cells with chromosomal deletion of *TIM23*, rescued by a *URA3*-containing plasmid carrying wild type *TIM23*, were transformed with plasmids carrying the *HIS3* gene as a selection marker and either a wild type copy of *TIM23* or *TIM23* mutant alleles. Transformants were selected on medium lacking Histidine (*Figure 1—figure supplement 1A*). The ability of *TIM23* mutants to complement Tim23 function was monitored by plasmid shuffle on 5-fluoroorotic acid (5-FOA)-containing medium (*Figure 1B*). Transformation was successful for all constructs, while 5-FOA selection showed that Tim23$^{G153L}$ exhibited a lethal phenotype as published previously (*Demishtein-Zohary et al., 2015*). *TIM23* mutants that grew on 5-FOA were subsequently analyzed for growth on fermentable (glucose) and non-fermentable (glycerol) carbon sources (*Figure 1C*). Four mutants, encoding Tim23$^{N150A}$, Tim23$^{L155A}$, Tim23$^{A156L}$ and Tim23$^{Y159A}$, exhibited a significant growth defect on non-fermentable media at 37°C, with Tim23$^{A156L}$ showing the strongest phenotype (*Figure 1C*).

To analyze whether the growth defects could be explained by changed channel characteristics of Tim23, we expressed wild type and mutant forms of Tim23 in *E. coli*. The proteins were purified from inclusion bodies to homogeneity, incorporated into preformed large unilamellar vesicles (LUVs) and subjected to single-channel planar lipid bilayer experiments (*Krüger et al., 2012*; *Montilla-Martinez et al., 2015*). Interestingly, in a wide screen for basic electrophysiological parameters we found that a number of mutants (*Tim23$^{N150A}$, Tim23$^{A156L}$, Tim23$^{Y159A}$*) that showed growth defects exhibited a significantly reduced reversal potential (*Figures 1D* and *3D* and *Figure 1—figure supplement 1B*), which translates to a severe reduction of the channels cation preference (*Figure 1E*), while other parameters remained unaffected (*Figure 1—figure supplement 1B–D*). The strongest reduction was observed for Tim23$^{N150A}$, where the selectivity dropped down to 33% of wild type level. A slightly weaker reduction in cation preference (between 50–70% of wild type level) was observed for Tim23$^{G153A}$, Tim23$^{A156G}$, Tim23$^{A156L}$, Tim23$^{Y159A}$ and Tim23$^{N160A}$. All residues with decreased selectivity are highly conserved between Tim23 in different species (*Figure 1—figure supplement 2*).

To analyze if the observed growth defects could be directly linked to altered channel characteristics or if they were secondary effects, we examined the integrity of the TIM23 complex in the inner membrane. Mitochondrial lysates of all mutants and wild type were analyzed for steady state protein levels of Tim23 (*Figure 2A*). Reduced levels of Tim23 were found for the mutants Tim23$^{L155A}$, Tim23$^{A156L}$, Tim23$^{Y159A}$ and Tim23$^{N160A}$ (*Figure 2A*, lanes 5, 7, 8 and 9). Tim23$^{L155A}$, Tim23$^{A156L}$, Tim23$^{Y159A}$ all showed impaired growth phenotypes, which might result from decreased Tim23 levels. To gain more insight into TIM23 complex integrity of the mutants we performed co-immunoprecipitation of wild type and all mutants using antibodies against Tim23 (*Figure 2B*). Interestingly, TIM23 and PAM subunits could be efficiently co-purified. The altered levels of some subunits (for example Tim17 and Tim50) can probably be attributed to decreased Tim23 levels in mitochondria. As an alternative approach, we analyzed TIM23 complex integrity of selected mutants by size exclusion chromatography. To this end, mitochondrial extracts carrying Tim23, Tim23$^{N150A}$, or Tim23$^{Y159A}$ were generated and subjected to chromatographic separation of protein complexes. In agreement with the results of the immunoisolation analyses, the TIM23 complex apparently remained intact and associated with the import motor (*Figure 2—figure supplement 1*).

Hence, after carefully testing the suitability of Tim23 mutants for subsequent analysis, Tim23$^{N150A}$ was the only mutation that led to impaired growth, decreased ion-selectivity and exhibited normal protein levels and complex assembly and was therefore further analyzed in *in organello* assays. Mitochondrial steady state levels of selected proteins were analyzed, that is, TIM23 complex components, PAM complex subunits and mitochondrial marker proteins (*Figure 2C*). Here, all protein levels in mitochondria from Tim23$^{N150A}$ expressing cells were unchanged compared to wild type.

To assess that the inner membrane potential was not affected in mitochondria containing Tim23$^{N150A}$, we tested the $\Delta\Psi$ *in organello*, using the membrane-permeable fluorophore DiSC$_3$(5) (*Figure 2D*). The measurements showed that $\Delta\Psi$ was not significantly altered in Tim23$^{N150A}$-expressing cells compared to the wild type control (*Figure 2D and E*). In agreement with this unchanged membrane potential, in single-channel measurements the IMS domain of Tim50 exhibited the same

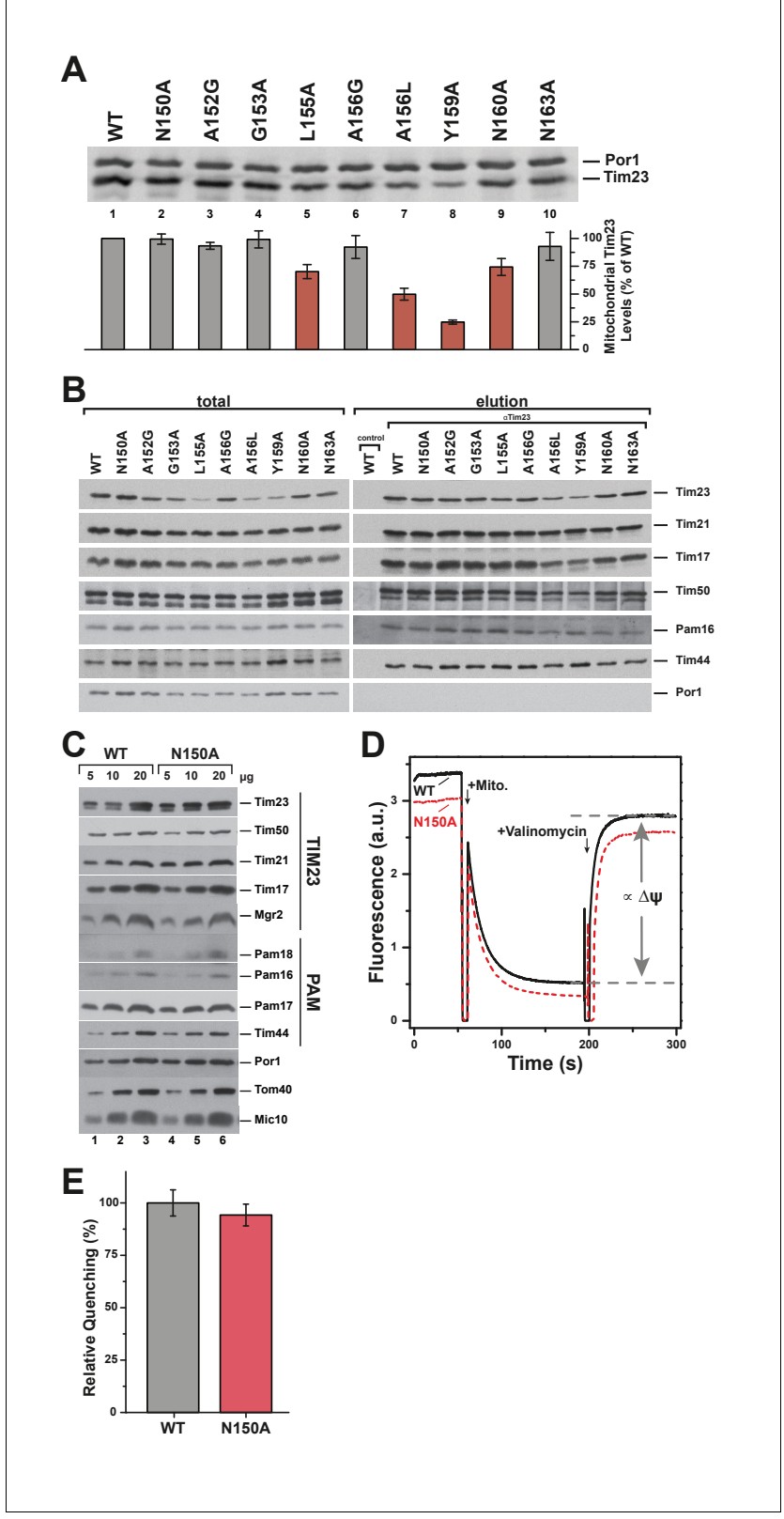

**Figure 2.** Pore-lining mutant Tim23[N150A] is properly expressed and integrated into TIM23 complexes. (**A**) Mitochondrial steady state levels of yeast expressing wild type Tim23 or mutants were assessed by Western blot analysis (upper) with decoration against Tim23 and Por1 (mitochondrial outer membrane). Protein levels were quantified using fluorescently labelled secondary antibodies in four independent experiments and normalized by

*Figure 2 continued on next page*

*Figure 2 continued*

mitochondrial Por1 levels. Significantly reduced levels are indicated in red. Error bars represent standard error of the mean. (B) TIM23 complex integrity and recruitment of PAM complex of wild type and Tim23 mutants was examined by co-immunoprecipitation of mitochondrial lysates using Tim23 antibodies. (C) Isolated mitochondria containing Tim23 or Tim23$^{N150A}$ were Western blotted and decorated against proteins of the TIM23 complex, the PAM complex, Por1 and Tom40 (both mitochondrial outer membrane) and Mic10 (mitochondrial inner membrane). (D) Membrane potential $\Delta\Psi$ was assessed by adding isolated mitochondria containing Tim23 (black solid) or Tim23$^{N150A}$ (red dashed) to the fluorophore DiSC$_3$(5), then dissipating $\Delta\Psi$ with valinomycin and determining the amount of quenching. Grey dashed lines and arrows indicate the parameter quantified in (E). (E) Relative fluorophore quenching as a measure of membrane potential $\Delta\Psi$ for Tim23 (grey) and Tim23$^{N150A}$ (red) was quantified (as depicted in D) in three independent experiments. Error bars represent standard error of the mean before normalization.

DOI: https://doi.org/10.7554/eLife.28324.006

The following figure supplement is available for figure 2:

**Figure supplement 1.** TIM23 complex characterization by size exclusion chromatography profiles.

DOI: https://doi.org/10.7554/eLife.28324.007

voltage-regulation on wild type and Tim23$^{N150A}$ that we reported before (*Figure 3E and F*) (*Meinecke et al., 2006*).

In our initial screen for altered electrophysiological characteristics, we found that specifically the ion-selectivity of Tim23$^{N150A}$ was decreased (*Figure 1E and 3D*). We next performed an in-depth analysis of this mutant form of the channel to confirm that no other channel parameters were affected. Wild type as well as Tim23$^{N150A}$ channels exhibited complex voltage-dependent gating patterns (*Figure 3A and B*). Both pores gated with the same main-conductance state of ~460 pS (at 250 mM KCl) and showed similar sub-conductance states of ~170 pS and ~60 pS (*Figure 3C*). Again Tim23$^{N150A}$ displayed a reduced reversal potential, while the wild type and mutant Tim23 showed the same voltage-dependent open probability (*Figure 3E and F*) and were efficiently voltage-regulated by Tim50$^{IMS}$ as published before (*Figure 3E and F*) (*Meinecke et al., 2006*). In summary, Tim23$^{N150A}$ is found in wild type levels in mitochondria, integrates properly into the TIM23 complex, and has no effect on the integrity of the inner membrane. In addition, it displays wild type-like channel characteristics except for a significantly reduced cation preference.

We next asked whether the reduced selectivity for cations impacted the import capabilities of the presequence translocase. To this end, isolated mitochondria were incubated with radiolabeled matrix proteins bearing typical, positively charged presequences: F$_1\beta$ (*Figure 4A*), a subunit of the F$_1$F$_O$-ATP synthase, Cox4 (*Figure 4B*), a subunit of the cytochrome *c* oxidase, and the model fusion proteins b$_2$(167)$_\Delta$-DHFR (*Figure 4C*) and b$_2$(220)-DHFR (*Figure 4D*) which is sorted into the inner membrane. The import reaction was stopped after 10, 20 or 30 min by dissipation of $\Delta\Psi$ and mitochondria were subsequently treated with Proteinase K to remove non-imported precursor proteins. Even at permissive temperature, quantified import efficiency in the linear phase revealed significant reductions for both types of imported substrates (*Figure 4E*), showing that Tim23$^{N150A}$ is clearly affected in protein import. Import experiments conducted at 37°C show the same trend with an even more pronounced reduction (*Figure 4—figure supplement 1*), while import experiments using the ADP/ATP carrier (AAC) and Cox12 revealed that other import pathways into mitochondria (TIM22 and MIA) were not impaired by the mutation (*Figure 4F and G*). In fact, a slightly increased import efficiency for AAC is frequently observed when transport along the TIM23 pathway is affected (*Geissler et al., 2002*; *Schulz et al., 2011*).

These observations led us to hypothesize that the reduced import capabilities of Tim23$^{N150A}$ were linked to the altered cation selectivity, which could be explained if selectivity defects lead to changed sensitivity of the mutant channel towards substrates. To test this, we analyzed the channel response of wild type Tim23 and Tim23$^{N150A}$ to presequences in single-channel experiments. As a substrate we used a peptide corresponding to the presequence of Cox4 (*Allison and Schatz, 1986*), a subunit of the cytochrome *c* oxidase, which is well characterized to study import processes and signal recognition biochemically (*Chacinska et al., 2005*; *Lytovchenko et al., 2013*; *Schulz et al., 2011*) and channel excitation electrophysiologically (*Lohret et al., 1997*; *Martinez-Caballero et al., 2007*; *Meinecke et al., 2006*; *Ramesh et al., 2016*; *Truscott et al., 2001*; *van der Laan et al., 2007*). The presequence peptide was titrated in increasing concentrations to the intermembrane

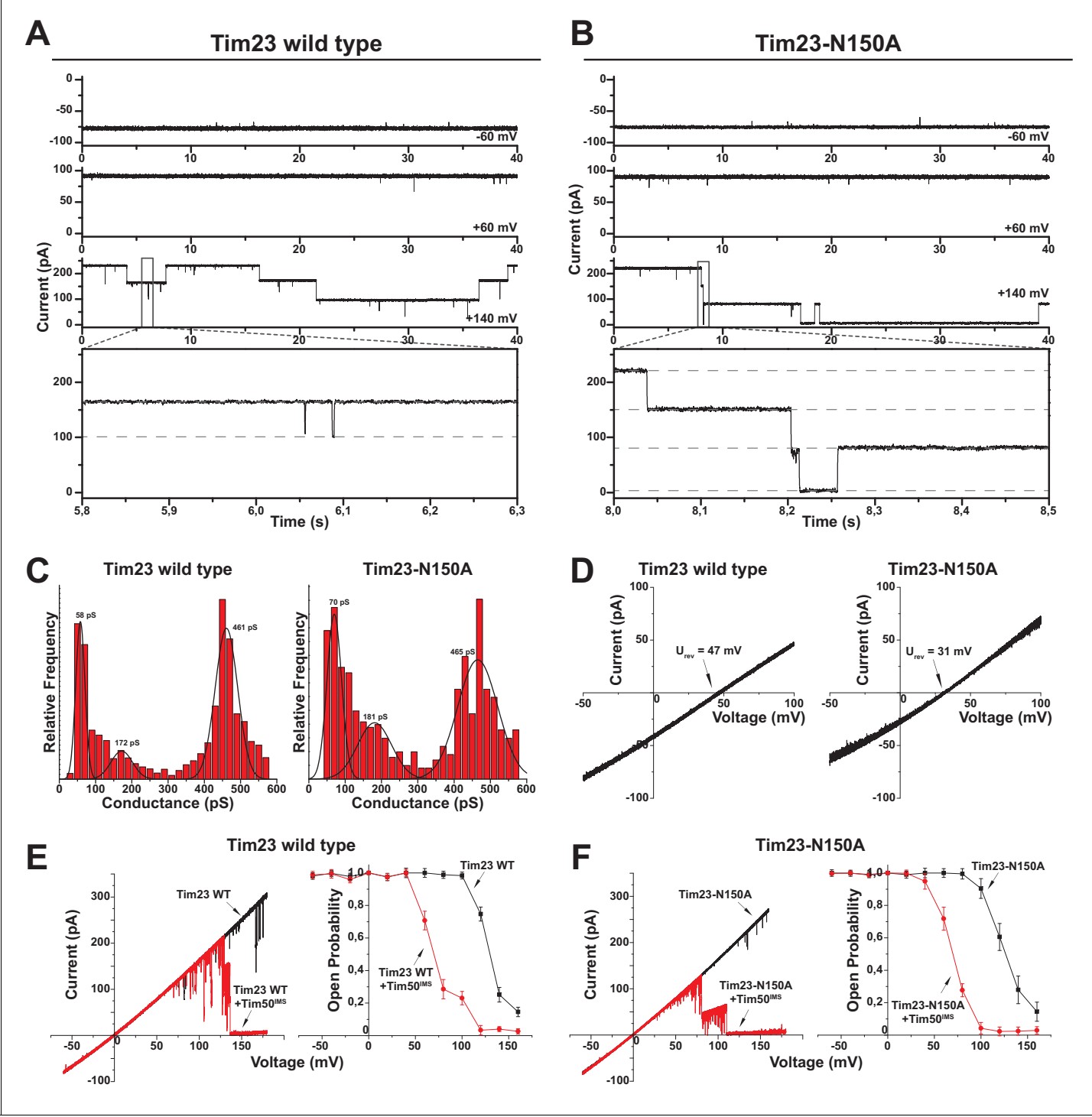

**Figure 3.** Tim23[N150A] displays reduced cation preference. (**A**)/(**B**) Tim23- (**A**) or Tim23[N150A]-containing (**B**) proteoliposomes were fused with planar lipid bilayers and single-channel activity was characterized by electrophysiological current recordings. (**C**) Gating event histograms for Tim23 (left) and Tim23[N150A] (right) were calculated from constant-voltage recordings (as depicted in A) with at least 2000 gating events each. The three most prominent classes of conductance changes were modeled with a Gaussian fit. (**D**) I-V curves at asymmetrical buffer conditions were recorded for Tim23 (left) and Tim23[N150A] (right) with indicated reversal potential $U_{rev}$ for 12.5-fold KCl-gradient. (**E**)/(**F**) I-V curves (left) and open probabilities (right) were determined for bilayer incorporated Tim23 (**E**) or Tim23[N150A] (**F**) before (black) and after (red) addition of 700 nM Tim50[IMS] to IMS-side of the channel. Error bars represent standard deviation (SD, n = 3).

DOI: https://doi.org/10.7554/eLife.28324.008

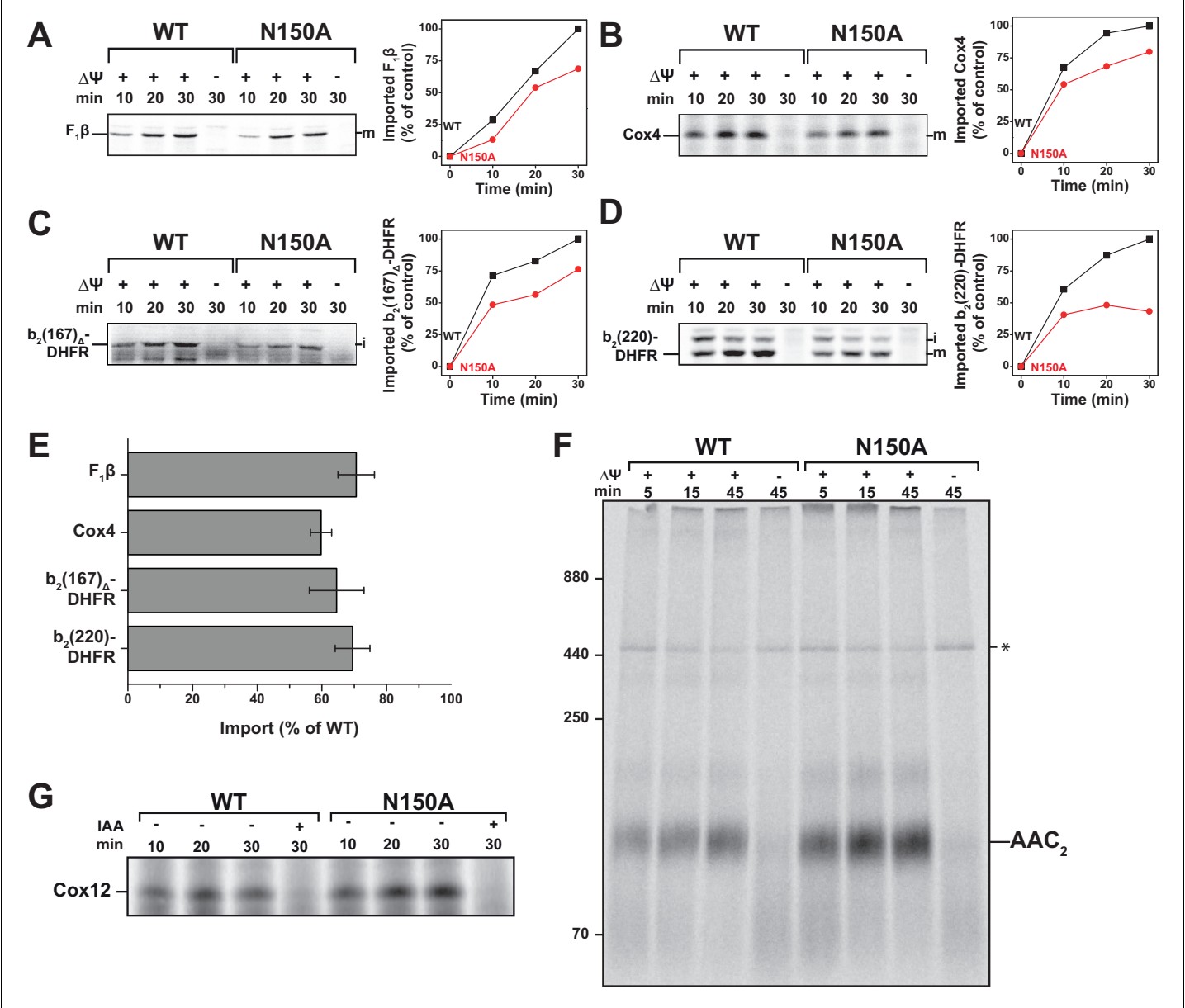

**Figure 4.** Tim23[N150A] exhibits significant import defects for various TIM23 substrates. (A–D) Import capability of wild type and tim23[N150A] mutant mitochondria was determined by incubating [$^{35}$S]-radiolabeled matrix destined precursors F$_1$β (A), Cox4 (B), b$_2$(167)$_\Delta$-DHFR (C) or the inner membrane sorted b$_2$(220)-DHFR (D) with isolated mitochondria for 10, 20 or 30 min. The import reactions were stopped by dissipating ΔΨ and subsequent Proteinase K (PK)-digest. Digital autoradiographs (left) were analyzed and quantified (right). Maximum import into wild type mitochondria was set to 100%. (E) Relative import efficiency after 15 min of import into mitochondria containing Tim23[N150A] was quantified for different substrates. Error bars represent standard error of the mean (SEM, n = 3). (F) Carrier import into Tim23[N150A]-containing mitochondria was assessed via ADP/ATP carrier (AAC) complex assembly by incubating [$^{35}$S]-radiolabeled AAC with isolated mitochondria for 15, 30 or 45 min. The import reaction was stopped by dissipating ΔΨ and subsequent PK-digest. Assembly of AAC dimer was monitored by BN-PAGE. (G) The MIA substrate Cox12 was [$^{35}$S]-radiolabeled and imported into Tim23[N150A]-containing mitochondria for 10, 20 or 30 min. The import reaction was stopped by addition of iodoacetamide (IAA) and subsequent PK-digest.

DOI: https://doi.org/10.7554/eLife.28324.009

The following figure supplement is available for figure 4:

**Figure supplement 1.** Import of TIM23-substrates at non-permissive temperatures.

DOI: https://doi.org/10.7554/eLife.28324.010

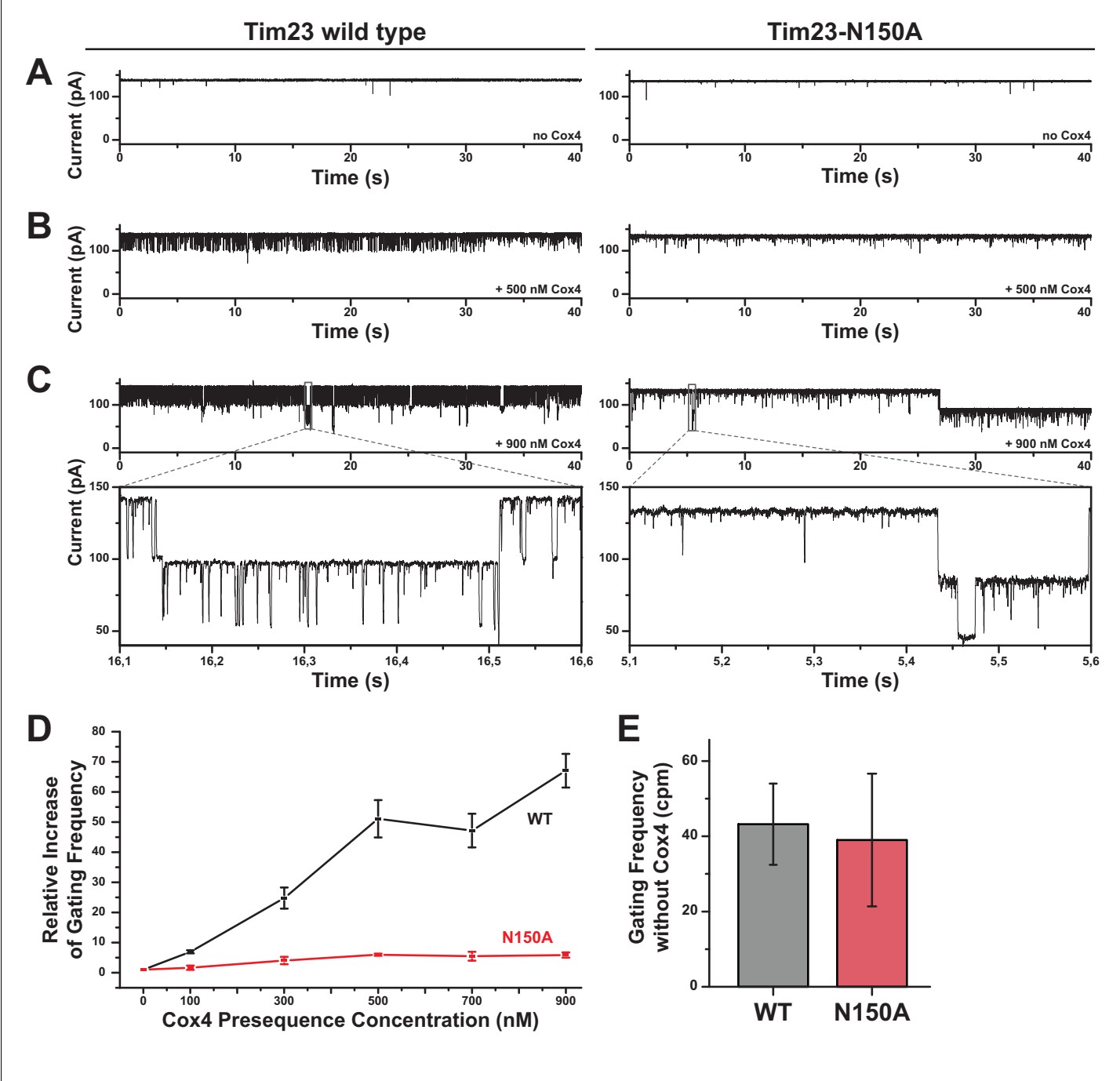

**Figure 5.** Increased gating frequency as a measure of channel response to presequence titration is heavily reduced for Tim23[N150A]. (**A**) Bilayer-incorporated Tim23 (left) or Tim23[N150A] (right) single-channel currents were recorded at +80 mV before addition of Cox4 presequence peptide. (**B**)/(**C**) Current recordings of wild type (left) and mutant (right) channels were performed after each titration step. (**D**) Channel response of Tim23 (black) and Tim23[N150A] (red) after addition of Cox4 presequence was quantified by counting gating events and calculating the relative increase in activity compared to unstimulated channels. Error bars represent standard deviation. (**E**) Absolute gating frequency of unstimulated Tim23 and Tim23[N150A] was determined from current recordings before presequence titration. Error bars represent standard deviation (SD, n = 3).

DOI: https://doi.org/10.7554/eLife.28324.011

space corresponding side of bilayer-incorporated wild type or Tim23[N150A] channels and current was recorded at constant holding potentials after each titration step. Tim23 reacted to higher holding potentials by partial or complete closing (*Figure 3A,B,E and F*), which could mask presequence-

induced activity increase or change channel behavior. Hence, we aimed to minimize such secondary effects by recording at lower holding potentials. We applied voltages of +80 mV, where the channel stayed primarily in an open state even during prolonged exposure but reacts to presequence activation. Wild type Tim23 showed a distinct activity increase, characterized by fast gating (flickering), after addition of Cox4 (*Figure 5A,B and C*). While Tim23[N150A] also responded with an increased gating frequency to Cox4 addition, the effect was drastically reduced in comparison to wild type Tim23. Where wild type Tim23 channels could be activated to a relative increase in gating frequency of factor 50 (*Figure 5D*, black curve), the gating frequency of Tim23[N150A] only changed by a factor 6 (*Figure 5D*, red curve), resulting in an 88% reduction of voltage-activated gating from wild type to mutant Tim23. To further prove that the relative reduction in gating frequency did not originate from e.g. a higher baseline of gating frequency of Tim23[N150A] compared to wild type Tim23, we analyzed absolute gating frequencies in the absence of substrate peptides (*Figure 5E*). Both proteins have near identical average gating frequencies in unstimulated conditions, excluding pre-activation effects.

In summary, our data show that a decreased selectivity of Tim23 leads to reduced substrate sensitivity, which explains the impaired protein import capacity of mitochondria expressing a Tim23 selectivity mutant.

## Discussion

Tim23, the eponymous core subunit of the TIM23 complex, was identified as the import channel for presequence carrying substrates more than 15 years ago (*Lohret et al., 1997*; *Truscott et al., 2001*). Although the basic channel characteristics have been described to some extend (*Martinez-Caballero et al., 2007*; *Meinecke et al., 2006*; *Truscott et al., 2001*), the physiological relevance of these parameters as well as the molecular mode of function of the import pore are still enigmatic. Due to lack of structural information the current understanding is little more than that at the heart of the TIM23 complex a water-filled pore facilitates the passage of preproteins across the inner membrane.

In this study, we investigated the molecular basis of Tim23's ion-selectivity to link biophysical properties of the import channel to its physiological function. We report the identification of pore-lining amino acids that contribute to the channels cation selectivity. Interestingly, we found that a number of different residues are involved in discriminating between ions. All amino acids detected to be important for channel selectivity are strictly conserved in evolution, suggesting an essential role for the selectivity in protein transport across the inner membrane. In recent studies these residues were successfully cross-linked to preproteins *in transit*, showing their accessibility and exposure to the channel lumen (*Alder et al., 2008*; *Malhotra et al., 2013*). Our results imply that selectivity is not necessarily provided by a confined restriction zone within the channel but rather by specific channel surface characteristics throughout the length of the pore, which facilitate the passage of certain ions over others. Such a mechanism was proposed for other large pores to explain their selective properties (*Im and Roux, 2002*; *Kutzner et al., 2011*). Even though these reports were mainly made for β-barrel proteins, the similarity of the electrophysiological properties between some β-barrel pores, like Tom40 or Sam50, and α-helical import pores, like Tim23 or Tim22, makes it appealing to speculate that the selectivity of these pores has a similar molecular nature. Most of the detected selectivity mutants in our study showed impaired growth. As an outlier Tim23[G153A] shows a mediocre reduction in selectivity while growth is relatively unaffected and we therefore hypothesize that a certain impairment of the selective properties can be compensated. We rigorously excluded mutant forms of Tim23 that showed decreased mitochondrial steady state levels or with a compromised TIM23 complex assembly and hence could not be used to link *in vitro* single-channel results with *in vivo* and *in organello* experiments. This led to the identification of Tim23[N150A], which is expressed and assembled as wild type Tim23. The potential across the inner membrane is unaffected in cells expressing Tim23[N150A]. In line with the uncompromised mitochondrial fitness, all channel parameters, especially Tim50[IMS] voltage-regulation, but selectivity and substrate sensitivity, are comparable with wild type Tim23. Importantly, mitochondria containing Tim23[N150A] channels showed significantly reduced import capacity for matrix proteins. We therefore conclude that the drastically reduced selectivity renders Tim23 channels insensitive towards positively charged substrate peptides, which in turn explains the reduced import rates of preproteins. The position of the amino acid

substitution N150A is found close to the beginning of transmembrane helix 2 and is therefore located at the matrix side of the channel. This suggests that the decreased substrate sensitivity characterized by an inactive, slowly gating channel in the presence of prepeptides cannot be explained by binding and activation of the substrate to the IMS side of the channel. Instead the substrate has to reach deep into the channel to be affected by a mutation at the matrix side. Similar to what was described for Tom40 (*Mahendran et al., 2012*), the active, fast-gating Tim23 is therefore likely a transport competent state of the channel, triggered by peptides in transit within the channel that are discriminated by the channels selective properties. A decreased selectivity of Tim23 leads to an inability to be activated and therefore directly to decreased import rates.

The strategy used in this study allowed us to link the biophysical properties of a protein translocase to its physiological function. While water-filled pores were identified at the heart of most organellar translocation complexes (*du Plessis et al., 2011*; *Harsman et al., 2010*; *Meinecke et al., 2016*; *Neupert and Herrmann, 2007*; *Schmidt et al., 2010*; *Sjuts et al., 2017*), only basic channel characteristics were described in most cases. A correlation between the fascinating electrophysiological properties of these large pores and their physiological function remained circumstantial evidence. It was for example speculated for almost 20 years, that the cation selectivity of the Tim23 channel is important for recognition or transport of positively charged presequences, though no direct evidence was reported. Interestingly, many translocation pores show partially similar electrophysiological properties. Although structurally diverse, Tim23, Tim22, Tom40 and Sam50 all display comparable channel diameters and a preference for cations. The molecular nature of these characteristics as well as their physiological importance will be highly important problems to tackle in the future.

## Materials and methods

### Protein expression and purification

*Sc*Tim23 wild type and mutants were expressed from the plasmid pET10N containing an N-Terminal His$_{10}$-Tag in *E. coli* strain BL21 (DE3). All mutants were generated from the wild type plasmid by site-directed mutagenesis. Inoculated cultures in LB-medium were grown to OD$_{600}$ ≈ 0.7, after expression (induced by 1 mM isopropyl-β-D-thiogalactopyranoside (IPTG), 37°C, 3 hr) cells were lysed and inclusion bodies were purified (*Meinecke et al., 2006*; *Tarasenko et al., 2017*). Inclusion bodies were then denatured by 8 M Urea, 150 mM NaCl, 10 mM Tris-HCl, 50 mM Imidazole, pH 8.0, applied to NiNTA-Agarose and eluted by the same buffer with 500 mM Imidazole. Isolated Tim23 was further subjected to size exclusion chromatography using a HiLoad 16/600 Superdex 75 column (GE Healthcare, NJ, USA) and single band purity was confirmed by SDS-PAGE.

The presequence-peptide Cox4 (MLSLRQSIRFFKPATRTLCSSRYLL) was purchased from JPT Peptide Technologies (DE) as N-terminal amine and C-terminal amide.

The IMS domain of Tim50 (aa 132–476) was recombinantly expressed and purified to single band purity as described elsewhere (*Geissler et al., 2002*; *Schulz et al., 2011*).

### Liposome preparation and protein incorporation

Lipids were purchased as L-α-Phosphatidylcholine (PC), L-α-Phosphatidylethanolamine (PE), L-α-Phosphatidylinositol (PI), L-α-Phosphatidylserine (PS) and Cardiolipin (CL) from Avanti Polar Lipids (AL, USA). The lipid mixture of 45:20:15:5:15 mol% PC:PE:PI:PS:CL in CHCl$_3$, closely resembling inner mitochondrial lipid composition (*van Meer et al., 2008*), was dried with a nitrogen stream and resuspended in 100 mM KCl, 10 mM MOPS-Tris, pH 7.0. Lipid suspension was thoroughly vortexed, subjected to seven freeze-thaw cycles and extruded through 200 nm membranes (Whatman plc, UK) to ensure unilamellarity and defined size distribution. Both liposomes and protein in urea were incubated with the mild detergent MEGA-9 (Glycon, DE) above CMC at 80 mM first separately then combined, at room temperature. Subsequently the liposome-protein-detergent mixture was subjected to dialysis against 5 L of liposome buffer to remove both urea and MEGA-9. Incorporation success was monitored by Histodenz flotation assay and sodium carbonate extractions as described elsewhere (*Barbot et al., 2015*).

## Electrophysiological experiments

Electrophysiological experiments with Tim23 were carried out using the planar lipid bilayer technique, described in detail before (Harsman et al., 2011; Reinhold et al., 2012). Briefly, Tim23-containing proteoliposomes were added next to the bilayer in the *cis* chamber to enable fusion of liposomes with the bilayer. Asymmetrical buffer conditions for osmotically-driven fusion were 250 mM KCl, 10 mM MOPS-Tris, pH 7.0 in the *cis* chamber and 20 mM KCl, 10 mM MOPS-Tris, pH 7.0, for a 12.5-fold KCl-gradient over the bilayer. The electric recordings were performed using two Ag/AgCl electrodes in glass tubes, embedded in a 2 M KCl agar-bridge to minimize junction potentials, with one electrode per chamber. The electrode in the *trans* chamber was the reference electrode as it was connected to the headstage (CV-5-1GU) of a Geneclamp 500B current amplifier (both Molecular Devices, CA, USA), with the cis-electrode acting as ground. Currents were digitized by a Digidata 1440A A/D converter and recorded using the software AxoScope 10.3 and Clampex 10.3 (all Molecular Devices). Analysis of the data was carried out using R-packages *stepR* (Hotz et al., 2013) and dbacf (Tecuapetla-Gómez and Munk, 2015) and OriginPro 8.5 (OriginLab, MA, USA). After incorporation of Tim23 into the lipid bilayer, symmetrical conditions were set by perfusion with 20x chamber volume of 250 mM KCl, 10 mM MOPS-Tris, pH 7.0. These symmetrical buffer conditions were used for constant-voltage recordings and current-voltage relations. Asymmetrical buffers identical to the fusion conditions were used for reversal potential measurements. Tim23 typically inserts unidirectionally into the bilayer, with the IMS-domain of Tim23 exposed to *trans*. For concentration-dependent quantification of gating events, the synthetic peptide representing the presequence of the TIM23-substrate cytochrome c oxidase subunit 4 was titrated to the *trans* chamber in increasing concentrations. After addition, the buffer in the chamber was stirred for 2 min and then rests for 2 min before current recordings start. Constant-voltage currents were recorded for one minute, all gating events were counted and used to determine the gating frequency, i.e. events per minute. The open probability was calculated by dividing the mean by the maximum current.

## Yeast growth and handling

All yeast strains were grown in YP medium (1% yeast extract, 2% peptone) with 2% glucose (YPD) or 3% glycerol (YPG) medium at 30°C. For plate growth test, synthetic medium containing 3% glycerol or 2% glucose was used. For generation of a Tim23 shuffling strain MB29 (Geissler et al., 2002), endogenous *TIM23* was replaced by homologous recombination with a *LYS2* cassette in a strain expressing *TIM23* from a *URA3* containing plasmid. Wild type and mutant Tim23 were expressed by cloning *TIM23* gene +1 kb upstream and downstream of the gene into pRS413. Point mutations were introduced by side-directed mutagenesis. After transformation of these plasmids into the shuffling strain, 5-Fluoroorotic acid (5-FOA) was used to select against *URA3* containing plasmids harboring wild type *TIM23*. For subsequent isolation of mitochondria, yeast cells were first grown in YPD medium at 30°C overnight, then diluted to $OD_{600}$ = 0.2 in YPG medium and continued to grow for 24 hr at 30°C. Cells were then transferred to a bigger culture at $OD_{600}$ = 0.2 in YPG and grown at 30°C for 16 hr, reaching a final $OD_{600}$ of 2–3. Isolation of mitochondria was handled as described before (Schendzielorz et al., 2017).

## Import of precursor proteins

For import of [$^{35}$S]-methionine labeled precursors into isolated mitochondria, proteins were translated using rabbit reticulocyte lysate (Promega, WI, USA). Reaction was performed in import buffer (250 mM sucrose, 10 mM MOPS/KOH pH 7.2, 80 mM KCl, 2 mM $KH_2PO_4$, 5 mM $MgCl_2$, 5 mM methionine and 3% fatty acid-free BSA) supplemented with 2 mM ATP and 2 mM NADH. To stop the import reaction, membrane potential was disrupted using final concentration of 1 μM valinomyin, 8 μM antimycin A and 20 μM oligomycin and samples were Proteinase K (PK, 20 μg/ml) treated for 10 min on ice. PK was inhibited with 2 mM phenylmethylsulphonyl fluoride (PMSF) for 10 min on ice; mitochondria were pelleted, washed with SEM (250 mM sucrose, 20 mM MOPS pH 7.2, 1 mM EDTA) and further analyzed by SDS-PAGE and autoradiography. Import and assembly of the ADP/ATP carrier protein (AAC) via TIM22 was performed using the standard protocol and further analyzed by Blue Native PAGE and autoradiography as described before (Schulz et al., 2011). Import of Cox12 via MIA was performed as described (Gornicka et al., 2014). Briefly, in addition to the standard protocol the reticulocyte lysate was diluted 1:2 in saturated ammonium sulfate (($NH_4$)$_2SO_4$)

and precipitated on ice. The pellet was resuspended in 8 M urea, 10 mM DTT, 30 mM MOPS, pH 7.2 and then added to mitochondria in import buffer without BSA. The Cox12 import reaction was stopped with 25 mM iodoacetamide (IAA) and PK.

Quantifications were performed using ImageQuant TL (GE Healthcare, NJ, USA) using a rolling ball background subtraction.

### Protein complex isolation

TIM23 complex isolation was carried out essentially as described (*Herrmann et al., 2001*). Briefly, mitochondria were resuspended to 1 mg/ml in solubilization buffer (20 mM Tris/HCl pH 7.4, 150 mM NaCl, 10% glycerol (w/v), 1 mM PMSF, 1% digitonin) and kept on ice for 20 min. Insoluble parts were removed by centrifugation at 14000 x g for 10 min and supernatant was incubated with Tim23-specific antibodies cross-linked to Protein A sepharose beads. After 30 min of binding on a rotating wheel at 4°C and 5x washing with 500 µl washing buffer (solubilization buffer with 0.3% digitonin), samples where eluted with 50 µl 0.1 mM glycine pH 2.8 (neutralized with 5 µl 1 M Tris base).

### Membrane potential measurements

Membrane potential was measured using 3,3'-dipropylthiadicarbocyanine iodide (DiSC$_3$(5)). Mitochondria were resuspended in buffer containing 600 mM sorbitol, 1% (wt/vol) BSA, 10 mM MgCl$_2$ and 20 mM KPi, pH 7.4 to a concentration of 166 µg/ml. Changes in fluorescence were assessed with a F-7000 fluorescence spectrophotometer (Hitachi, JP) at room temperature with excitation at 622 nm, emission at 670 nm and slits of 5 nm. Components were added to the cuvette in the fowling order: 500 µl of buffer, DiSC$_3$(5), 83 µg of mitochondria, 1 µM valinomycin. To compare relative differences in membrane potential, the difference in fluorescence before and after addition of valinomycin was used.

### Complex characterization by size exclusion chromatographie

Mitochondria were solubilized as described for protein complex isolation, with 2 mg/ml instead of 1 mg/ml, and 50 µl were loaded to a Superose 6 increase 3.2/300 and eluted in solubilization buffer containing 0.3% digitonin. The first 1250 µl were discarded, the following 750 µl were collected in 50 µl fractions and subjected to SDS-PAGE and Western blot. Quantifications were performed using ImageQuant TL (GE Healthcare, NJ, USA) using a rolling ball background subtraction.

## Acknowledgements

We thank Robert Rucktäschel for technical assistance with size exclusion chromatography. The work was supported by the Deutsche Forschungsgemeinschaft SFB1190, P12 (MM), SFB860, B1 (PR), FOR 1905, TP1 (ND), the Boehringer Ingelheim Foundation (FR), the Max Planck Society (PR) and the Ph. D program 'Molecular Biology' – International Max Planck Research School and the Göttingen Graduate School for Neurosciences and Molecular Biosciences (GGNB; DFG grant GSC 226/1 (ABS)).

## Additional information

### Funding

| Funder | Grant reference number | Author |
| --- | --- | --- |
| Deutsche Forschungsgemeinschaft | SFB1190 | Michael Meinecke |
| Deutsche Forschungsgemeinschaft | SFB860 | Peter Rehling |
| Deutsche Forschungsgemeinschaft | FOR1905 | Niels Denkert |

The funders had no role in study design, data collection and interpretation, or the decision to submit the work for publication.

## Author contributions
Niels Denkert, Data curation, Formal analysis, Investigation, Methodology, Writing—original draft; Alexander Benjamin Schendzielorz, Formal analysis, Investigation, Methodology; Mariam Barbot, Conceptualization, Investigation; Lennart Versemann, Frank Richter, Investigation; Peter Rehling, Conceptualization, Supervision, Writing—review and editing; Michael Meinecke, Conceptualization, Data curation, Supervision, Funding acquisition, Writing—original draft

## Author ORCIDs
Niels Denkert ⓘD http://orcid.org/0000-0002-2551-0360
Alexander Benjamin Schendzielorz ⓘD http://orcid.org/0000-0003-3360-9130
Peter Rehling ⓘD http://orcid.org/0000-0001-5661-5272
Michael Meinecke ⓘD http://orcid.org/0000-0003-1414-6951

## Decision letter and Author response
Decision letter https://doi.org/10.7554/eLife.28324.013
Author response https://doi.org/10.7554/eLife.28324.014

# Additional files
## Supplementary files
• Transparent reporting form
DOI: https://doi.org/10.7554/eLife.28324.012

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
