## [Decision Letter]

Thank you for submitting your article "Cation selectivity of the presequence translocase channel Tim23 is crucial for efficient protein import" for consideration by *eLife*. Your article has been favorably evaluated by Anna Akhmanova (Senior Editor) and three reviewers, one of whom, Nikolaus Pfanner, is a member of our Board of Reviewing Editors. The following individual involved in review of your submission has agreed to reveal his identity: Andre Schneider (Reviewer #2).

The reviewers have discussed the reviews with one another and the Reviewing Editor has drafted this decision to help you prepare a revised submission.

Summary:

Protein translocation across or into the mitochondrial inner membrane is mediated by an N-terminal presequence in conjunction with the TIM23 complex. While its central component, Tim23 is believed to form a protein-conducting channel, the molecular mechanisms of preprotein translocation through the channel remain largely unknown. Denkert et al. characterize a panel of nine Tim23 variants whose second transmembrane helices contain point mutations. The growth on non-fermentable carbon sources of four of these mutants is reduced at elevated temperature. Six mutants exhibit a reduction in cation selectivity when analyzed by electrophysiological methods. The mutant N150A, which showed the greatest reduction and also exhibited a growth phenotype, was selected for further analysis. A very detailed electrophysiological analysis shows that besides the cation selectivity no other channel parameters were altered in the N150A mutant. This study establishes a link between the reduced import activity observed in isolated mitochondria and the reduced presequence-induced gating caused by the altered cation selectivity of the recombinant Tim23 import channel.

This is a highly interesting study that combines in organello and biophysical approaches. The data is of high quality and provides insight into the physiological functioning of the essential Tim23 subunit.

Essential revisions:

1) Figure 1 or 2 should be complemented with BN-PAGE analysis, i.e. mitochondria from mutants grown on glycerol. It is important to see if the Tim23 mutations are influencing overall assembly of protein into the TIM23 complex (currently only shown for N150A in Figure 2 via pull-down).

2) The authors studied the import capacity of Tim23N150A by importing one natural presequence precursor and a model fusion protein. Since the natural presequence protein F_1_β showed a milder import defect, the authors should provide further import data of other native presequence-containing proteins and should check the import of precursor proteins destined for other mitochondrial import pathways, e.g. TIM22, SAM, MIA, to validate this observation is specific and general mitochondrial fitness is not being affected. In addition, the lateral sorting of presequence-containing precursor proteins into the inner membrane should also be studied to gain more information about the import deficiency in this mutant and its relation to the cation selectivity of Tim23.

3) Figure 2 claims that L155A (lane 5) and Y159A (lane 8) are reduced and hence were not looked at further. These mutants do not appear significantly reduced and the justification for why they have not been looked at further is weak. Quantification would enhance the author's arguments, but at the very least mitochondria could be analyzed as has been done in Figure 2 for N150A and A156L. This would make the story and flow of the manuscript clearer.

Likewise in Figure 2 the A156L mutant does not seem reduced in the whole cell extracts (panel 2A) but is reduced in isolated mitochondria (Figure 2). Are all of these mutant proteins exclusively targeted to mitochondria? The rationale for dropping analysis of A156L is also unclear. The story would be strengthened if it included another mutant so the case for why they didn't proceed with analysis of this mutant needs to be enhanced and backed up with some experimental information. For example, is the mutant not assembling into TIM23, or is the membrane potential affected etc.?

4) The reviewer disagrees with the statement "Notably there is a good accordance between the mutants that show a growth defects and those that displayed reduced selectivity properties". The mutant with the strongest growth phenotype A156L, for example, does not show the greatest reduction in cation preference. Moreover, G153A grows normally but shows reduced cation preference, whereas L155A has a growth phenotype but shows a normal cation preference. It would be helpful if the authors could discuss this in more detail.

---

## [Author Response]

*Essential revisions:*

*1) Figure 1 or 2 should be complemented with BN-PAGE analysis, i.e. mitochondria from mutants grown on glycerol. It is important to see if the Tim23 mutations are influencing overall assembly of protein into the TIM23 complex (currently only shown for N150A in Figure 2 via pull-down).*

As suggested, we performed additional experiments that strengthen our initial results from the mutant screening and provide new data in Figure 2 to address this issue. We now show mitochondrial steady state levels of Tim23 including a quantification that clearly shows decreased Tim23 levels in all mutants we previously excluded. Despite all efforts, we could not provide a reliable BN-PAGE analysis of TIM23. It appears that this is due to a lack of suitable antibodies. As an alternative approach, we performed co-immuno precipitations of Tim23 in all strains and analysed if TIM23 and PAM subunits can be co-purified (new Figure 2). Additionally, we performed size exclusion chromatography with subsequent Western blot analysis to examine TIM23 complex integrity in selected mutants (Figure 2—figure supplement 1).

*2) The authors studied the import capacity of Tim23N150A by importing one natural presequence precursor and a model fusion protein. Since the natural presequence protein F_1_β showed a milder import defect, the authors should provide further import data of other native presequence-containing proteins and should check the import of precursor proteins destined for other mitochondrial import pathways, e.g. TIM22, SAM, MIA, to validate this observation is specific and general mitochondrial fitness is not being affected. In addition, the lateral sorting of presequence-containing precursor proteins into the inner membrane should also be studied to gain more information about the import deficiency in this mutant and its relation to the cation selectivity of Tim23.*

To address this point, we performed additional import experiments using a number of different substrates, which are now shown in the revised version of Figure 3. The tested substrates showed similar decreased import rates, in agreement with our initial results. In detail, we performed import experiments with matrix targeted substrates F_1_β, Cox4 and b_2_(167)_Δ_-DHFR and with a sorted inner membrane substrate b_2_(220)-DHFR. All substrates show similarly decreased import in Tim23^N150A^ mitochondria (revised Figure 4). As requested, we assessed if other import routes are affected in Tim23^N150A^ mitochondria. As shown in Figure 4 neither the import of AAC (TIM22 substrate) nor Cox12 (MIA substrate) are affected in mutant mitochondria.

3) Figure 2 claims that L155A (lane 5) and Y159A (lane 8) are reduced and hence were not looked at further. These mutants do not appear significantly reduced and the justification for why they have not been looked at further is weak. Quantification would enhance the author's arguments, but at the very least mitochondria could be analyzed as has been done in Figure 2 for N150A and A156L. This would make the story and flow of the manuscript clearer.

*Likewise in Figure 2 the A156L mutant does not seem reduced in the whole cell extracts (panel 2A) but is reduced in isolated mitochondria (Figure 2). Are all of these mutant proteins exclusively targeted to mitochondria? The rationale for dropping analysis of A156L is also unclear. The story would be strengthened if it included another mutant so the case for why they didn't proceed with analysis of this mutant needs to be enhanced and backed up with some experimental information. For example, is the mutant not assembling into TIM23, or is the membrane potential affected etc.?*

To address this point experimentally we purified mitochondria from different strains, analyzed steady state levels of Tim23 by Western blotting, and quantified the amount of protein from four independent experiments (revised Figure 2). The new results are in line with our initial observation. We agree that a second suitable mutant would have been desirable, nonetheless detailed analysis of mitochondria carrying the varying Tim23 mutants didn’t provide a suitable strain. We repeated the steady state analysis of Tim23 in the various mitochondria and, as stated before, the levels of Tim23^L155A^, Tim23^A156L^, Tim23^Y159A^ and Tim23^N160A^ were significantly reduced and therefore excluded from further experiments. To further support and extend our selection process we performed immunoprecipitation analyses, similar to what was shown before for Tim23^N150A^ (old Figure 3), for all mutants. This analysis clearly shows that Tim23^N150A^ is the only mutant with reduced growth that displays WT levels of all TIM23 complex proteins tested (Figure 1 and Figure 2).

*4) The reviewer disagrees with the statement "Notably there is a good accordance between the mutants that show a growth defects and those that displayed reduced selectivity properties". The mutant with the strongest growth phenotype A156L, for example, does not show the greatest reduction in cation preference. Moreover, G153A grows normally but shows reduced cation preference, whereas L155A has a growth phenotype but shows a normal cation preference. It would be helpful if the authors could discuss this in more detail.*

We acknowledge the confusion and have deleted this statement. As suggested, we have extended the Discussion and included a paragraph addressing the raised issue.